# ThreeDWorld: A Platform for Interactive Multi-Modal Physical Simulation

**Chuang Gan**[1], **Jeremy Schwartz**[2], **Seth Alter**[2], **Damian Mrowca**[4], **Martin Schrimpf**[2],
**James Traer**[2], **Julian De Freitas**[3], **Jonas Kubilius**[2], **Abhishek Bhandwaldar**[1], **Nick Haber**[4],
**Megumi Sano**[4], **Kuno Kim**[4], **Elias Wang**[4], **Michael Lingelbach**[4], **Aidan Curtis**[2],
**Kevin Feigelis**[4], **Daniel M. Bear**[4], **Dan Gutfreund**[1], **David Cox**[1], **Antonio Torralba**[2],
**James J. DiCarlo**[2], **Joshua B. Tenenbaum**[2], **Josh H. McDermott**[2], **Daniel L.K. Yamins**[4]
[1] MIT-IBM Watson AI Lab, [2] MIT, [3] Harvard University, [4] Stanford University

www.threedworld.org

## Abstract

We introduce ThreeDWorld (TDW), a platform for interactive multi-modal physical simulation. TDW enables simulation of high-fidelity sensory data and physical interactions between mobile agents and objects in rich 3D environments. Unique properties include: real-time near-photo-realistic image rendering; a library of objects and environments, and routines for their customization; generative procedures for efficiently building classes of new environments; high-fidelity audio rendering; realistic physical interactions for a variety of material types, including cloths, liquid, and deformable objects; customizable "agents" that embody AI agents; and support for human interactions with VR devices. TDW's API enables multiple agents to interact within a simulation and returns a range of sensor and physics data representing the state of the world. We present initial experiments enabled by TDW in emerging research directions in computer vision, machine learning, and cognitive science, including multi-modal physical scene understanding, physical dynamics predictions, multi-agent interactions, models that 'learn like a child', and attention studies in humans and neural networks.

## 1 Introduction

A longstanding goal of research in artificial intelligence is to engineer machine agents that can interact with the world, whether to assist around the house, on a battlefield, or in outer space. Such AI systems must learn to perceive and understand the world around them in physical terms in order to be able to manipulate objects and formulate plans to execute tasks. A major challenge for developing and benchmarking such agents is the logistical difficulty of training an agent. Machine perception systems are typically trained on large data sets that are laboriously annotated by humans, with new tasks often requiring new data sets that are expensive to obtain. And robotic systems for interacting with the world pose a further challenge – training by trial and error in a real-world environment is slow, as every trial occurs in real-time, as well as expensive and potentially dangerous if errors cause damage to the training environment. There is thus growing interest in using simulators to develop and benchmark embodied AI and robot learning models [23, 42, 33, 35, 44, 10, 38, 45, 7].

World simulators could in principle greatly accelerate the development of AI systems. With virtual agents in a virtual world, training need not be constrained by real-time, and there is no cost to errors (e.g. dropping an object or running into a wall). In addition, by generating scenes synthetically, the researcher gains complete control over data generation, with full access to all generative parameters, including physical quantities such as mass that are not readily apparent to human observers and therefore difficult to label. Machine perceptual systems could thus be trained on tasks that are not

35th Conference on Neural Information Processing Systems (NeurIPS 2021) Track on Datasets and Benchmarks.

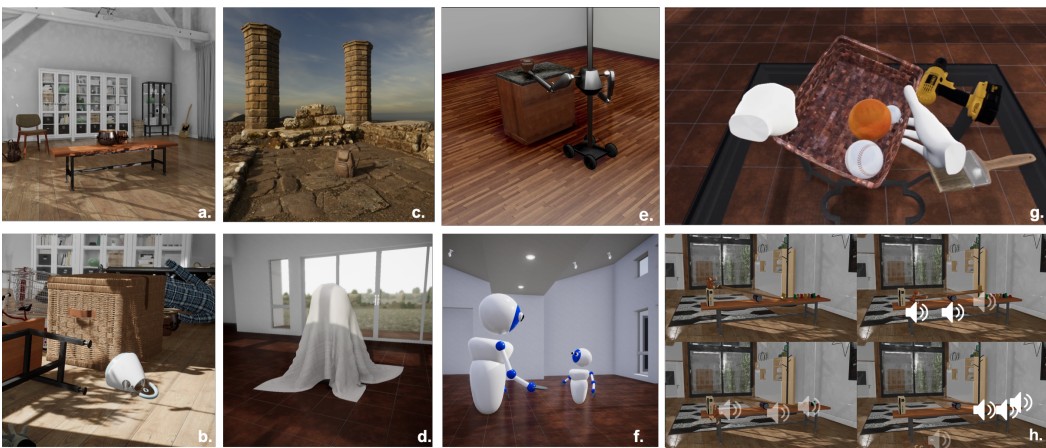

Figure 1: TDW's general, flexible design supports a broad range of use-cases at a high level of multi-modal fidelity: a-c) Indoor and outdoor scene rendering; d) Advanced physics – cloth draping over a rigid body; e) Robot agent picking up object; f) Multi-agent scene – "parent" and "baby" avatars interacting; g) Human user interacting with virtual objects in VR; h) Multi-modal scene – speaker icons show playback locations of synthesized impact sounds.

well suited to the traditional approach of massively annotated real-world data. A world simulator can also in principle simulate a wide variety of environments, which may be crucial to avoid overfitting.

The past several years have seen the introduction of a variety of simulation environments tailored to particular research problems in embodied AI, scene understanding, and physical inference. Simulators have stimulated research in navigation (*e.g.,* Habitat [35], iGibson [43]), robotic manipulation (*e.g.,* Sapien [45]), and embodied learning (*e.g.,* AI2Thor [23]). The impact of these simulators is evident in the many challenges they have enabled in computer vision and robotics. Existing simulators each have various strengths, but because they were often designed with specific use cases in mind, each is also limited in different ways. In principle a system could be trained to see in one simulator, to navigate in another and to manipulate objects in a third. However, switching platforms is costly for the researcher. We saw the need for a single simulation environment that is broadly applicable to the science of intelligence, by combining rich audio-visual rendering, realistic physics, and flexibility.

ThreeDWorld (TDW) is a general-purpose virtual world simulation platform that supports multi-modal physical interactions between objects and agents. TDW was designed to accommodate a range of key domains in AI, including perception, interaction, and navigation, with the goal of enabling training in each of these domains within a single simulator. It is differentiated from existing simulation environments by combining high-fidelity rendering for both video and audio, realistic physics, and a single flexible controller.

In this paper, we describe the TDW platform and its key distinguishing features, as well as several example applications that illustrate its use in AI research. These applications include: 1) A learned visual feature representation, trained on a TDW image classification dataset comparable to ImageNet, transferred to fine-grained image classification and object detection tasks; 2) A synthetic dataset of impact sounds generated via TDW's audio impact synthesis and used to test material and mass classification, using TDW's ability to handle complex physical collisions and non-rigid deformations; 3) An agent trained to predict physical dynamics in novel settings; 4) Sophisticated multi-agent interactions and social behaviors enabled by TDW's support for multiple agents; 5) Experiments on attention comparing human observers in VR to a neural network agent.

A download of TDW's full codebase and documentation is available at: https://github.com/threedworld-mit/tdw; the code for creating the datasets described below are available at: TDW-Image, TDW-Sound, and TDW-Physics.

**Related Simulation Environments** TDW is distinguished from many other existing simulation environments in the diversity of potential use cases it enables. A summary comparison of TDW's features to those of existing environments is provided in Table 1. These environments include

AI2-THOR[23], HoME[42], VirtualHome[33], Habitat[35], Gibson[44], iGibson [43], Sapien [45] PyBullet [10], MuJuCo [38], and Deepmind Lab [7].

TDW is unique in its support of: a) Real-time near-photorealistic rendering of both indoor and outdoor environments; b) A physics-based model for generating situational sounds from object-object interactions (Fig. 1h); c) Procedural creation of custom environments populated with custom object configurations; d) Realistic interactions between objects, due to the unique combination of high-resolution object geometry and fast-but-accurate high-resolution rigid body physics (denoted "$R^+$" in Table 1); e) Complex non-rigid physics, based on the NVIDIA Flex engine; f) A range of user-selectable embodied agent agents; g) A user-extensible model library.

Table 1: Comparison of TDW's capabilities with those of related virtual simulation frameworks.

| Platform | Scene (I,O) | Physics (R/$R^+$,S,C,F) | Acoustic (E,P) | Interaction (D,A,H) | Models (L,E) |
|---|---|---|---|---|---|
| Deepmind Lab [7] | | | | D, A | |
| MuJuCo [38] | | $R^+$, C, S | | D, A | |
| PyBullet [10] | | $R^+$, C, S | E | D, A | |
| HoME [42] | | R | E | | |
| VirtualHome [33] | I | | | D, A | |
| Gibson [44] | I | | | | |
| iGibson [43] | I | $R^+$ | | D, A ,H | L |
| Sapien [45] | I | $R^+$ | | D, A | L |
| Habitat [35] | I | | E | | |
| AI2-THOR [23] | I | R | | D | L |
| ThreeDWorld | I, O | $R^+$, C, S, F | E, P | D, A, H | L, E |

**Summary:** Table 1 shows TDW differs from these frameworks in its support for different types of:

- Photorealistic scenes: indoor (I) and outdoor (O)
- Physics simulation: just rigid body (R) or improved fast-but-accurate rigid body ($R^+$), soft body (S), cloth (C) and fluids (F)
- Acoustic simulation: environmental (E) and physics-based (P)
- User interaction: direct API-based (D), agent-based (A) and human-centric using VR (H)
- Model library support: built-in (L) and user-extensible (E)

## 2 ThreeDWorld Platform

### 2.1 Design Principles and System Overview

**Design Principles.** Our core contribution is to integrate several existing real-time advanced physics engines into a framework that can also produce high-quality visual and auditory renderings. In making this integration, we followed three design principles:

- The integration should be flexible. That is, users should be able to easily set up a wide variety of physical scenarios, placing any type of object at any location in any state, with controllable physical parameters. This enables researcher to create physics-related benchmarks with highly variable situations while also being able to generate near-photorealistic renderings of those situations.
- The physics engines should cover a wide variety of object interactions. We achieve this aim by seamlessly integrating PhysX (a good rigid-body simulator) and Nvdia Flex (a state-of-the-art multi-material simulator for non-rigid and rigid-non-rigid interactions).
- There should be a large library of high-quality assets with good physics colliders as well as realistic rigid and non-rigid material types, to allow users to take advantage of the power of the physics engines and easily be able to produce interesting and useful physical scenes.

**System Overview.** The TDW simulation consists of two basic components: (i) the **Build**, a compiled executable running on the Unity3D Engine, which is responsible for image rendering, audio synthesis and physics simulations; and (ii) the **Controller**, an external Python interface to communicate with the build. Users can define their own tasks through it, using an API comprising over 200 commands. Running a simulation follows a cycle in which: 1) The controller sends **commands** to the build; 2) The build executes those commands and sends **simulation output data** back to the controller. Unlike

other simulation platforms, TDW's API commands can be combined into lists and sent to the build within a single time step, allowing the simulation of arbitrarily complex behavior. Researchers can use this core API as a foundation on which to build higher-level, application-specific API "layers" that dramatically reduce development time and enable widely divergent use cases.

## 2.2 Photo-realistic Rendering

TDW uses Unity's underlying game-engine technology for image rendering, adding a custom lighting approach to achieve near-photorealistic rendering quality for both indoor and outdoor scenes.

**Lighting Model.** TDW uses two types of lighting; a single light source simulates direct light coming from the sun, while indirect environment lighting comes from "skyboxes" that utilize High Dynamic Range (HDRI) images. For details, see Fig 1(a-c) and the Supplement. Additional post-processing is applied to the virtual camera including exposure compensation, tone mapping and dynamic depth-of-field (examples).

**3D Model Library.** To maximize control over image quality we have created a library of 3D model "assets" optimized from high-resolution 3D models. Using Physically-Based Rendering (PBR) materials, these models respond to light in a physically correct manner. The library contains around 2500 objects spanning 200 categories organized by Wordnet synset, including furniture, appliances, animals, vehicles, and toys etc. Our material library contains over 500 materials across 10 categories, many scanned from real world materials.

**Procedural Generation of New Environments.** In TDW, a run-time virtual world, or "scene", is created using our 3D model library assets. Environment models (interior or exterior) are populated with object models in various ways, from completely procedural (i.e. rule-based) to thematically organized (i.e. explicitly scripted). TDW places no restrictions on which models can be used with which environments, which allows for unlimited numbers and types of scene configurations.

## 2.3 High-fidelity Audio Rendering

Multi-modal rendering is an unique aspect of TDW, and our audio engine provides both physics-driven impact sound generation, and reverberation and spatialized sound simulation.

**Generation of Impact Sounds.** TDW's includes PyImpact, a Python library that uses modal synthesis to generate impact sounds [39]. PyImpact uses information about physical events such as material types, as well as velocities, normal vectors and masses of colliding objects to synthesize sounds that are played at the time of impact (examples). This "round-trip" process is real-time. Synthesis is currently being extended to encompass scraping and rolling sounds [1].

**Environmental Audio and Reverberation.** For sounds placed within interior environments, TDW uses a combination of Unity's built-in audio and Resonance Audio's 3D spatialization to provide real-time audio propagation, high-quality simulated reverberation and directional cues via head-related transfer functions. Sounds are attenuated by distance and can be occluded by objects or environment geometry. Reverberation automatically varies with the geometry of the space, the virtual materials applied to walls, floor and ceiling, and the percentage of room volume occupied by solid objects (*e.g.,* furniture).

## 2.4 Physical Simulation

In TDW, object behavior and interactions are handled by a physics engine. TDW now integrates two physics engines, supporting both rigid-body physics and more advanced soft-body, cloth and fluid simulations.

**Rigid-body physics.** Unity's rigid body physics engine (PhysX) handles basic physics behavior involving collisions between rigid bodies. To achieve accurate but efficient collisions, we use the powerful V-HACD algorithm [28] to compute "form-fitting" convex hull colliders around each library object's mesh, used to simplify collision calculations (see Figure 2). In addition, an object's

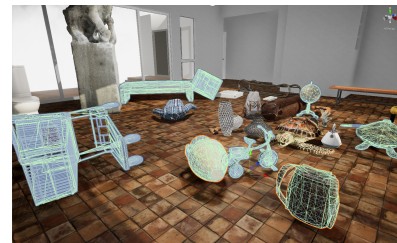

Figure 2: Green outlines around objects indicate auto-computed convex colliders for fast but accurate rigid-body physics.

mass is automatically calculated from its volume and material density upon import. However, using API commands it is also possible to dynamically adjust mass or friction, as well as visual material appearance, on a per-object basis enabling potential disconnection of visual appearance from physical behavior (e.g. objects that look like concrete but bounce like rubber).

**Advanced Physics Simulations.** TDW's second physics engine – Nvidia Flex – uses a particle-based representation to manage collisions between different object types. TDW supports rigid body, soft body (deformable), cloth and fluid simulations Figure 1(d). This unified representation helps machine learning models use underlying physics and rendered images to learn a physical and visual representation of the world through interactions with objects in the world.

## 2.5   Interactions and Agents

TDW provides three paradigms for interacting with 3D objects: 1) Direct control of object behavior using API commands. 2) Indirect control through an embodiment of an AI agent. 3) Direct interaction by a human user, in virtual reality (VR).

**Direct Control.** Default object behavior in TDW is completely physics-based via commands in the API; there is no scripted animation of any kind. Using physics-based commands, users can move an object by applying an impulse force of a given magnitude and direction.

**Agents.** The embodiment of AI agents come in several types:
- Disembodied cameras for generating first-person rendered images, segmentation and depth maps.
- Basic embodied agents whose avatars are geometric primitives such as spheres or capsules that can move around the environment and are often used for algorithm prototyping.
- More complex embodied avatars with user-defined physical structures and associated physically-mapped action spaces. For example, TDW's Magnebot is a complex robotic body, fully physics-driven with articulated arms terminating in 9-DOF end-effectors (Fig. 1e). By using commands from its high-level API such as **reach_for(target position)** and **grasp(target object)**, Magnebot can be made to open boxes or pick up and place objects. In addition, as a first step towards sim2real transfer, researchers can also import standard URDF robot specification files into TDW and use actual robot types such as Fetch, Sawyer or Baxter as embodied agents.

Agents can move around the environment while responding to physics, using their physics-driven articulation capabilities to change object or scene states, or can interact with other agents within a scene (Fig. 1f).

**Human Interactions with VR devices.** TDW also supports users interacting directly with 3D objects using VR. Users see a 3D representation of their hands that tracks the actions of their own hands (Fig. 1g). Using API commands, objects are made "graspable" such that any collision between object and virtual hands allows the user to pick it up, place it or throw it (example). This functionality enables the collection of human behavior data, and allows humans to interact with avatars.

## 3   Example Applications

### 3.1   Visual and Sound Recognition Transfer

We quantitatively examine how well feature representations learned using TDW-generated images and audio data transfer to real world scenarios.

**Visual recognition transfer** We generated a TDW image classification dataset comparable in size to ImageNet; 1.3M images were generated by randomly placing one of TDW's 2,000 object models in an environment with random conditions (weather, time of day) and taking a snapshot while pointing the randomly positioned virtual camera at the object ( **Details in Supplement**).

Table 2: Visual representations transfer for fine-grained image classifications.

| Dataset | Aircraft | Bird | Car | Cub | Dog | Flower | Food | Mean |
|---------|----------|------|------|------|------|--------|------|------|
| ImageNet | 0.74 | 0.70 | 0.86 | 0.72 | 0.72 | 0.92 | 0.83 | 0.78 |
| SceneNet | 0.06 | 0.43 | 0.30 | 0.27 | 0.38 | 0.62 | 0.77 | 0.40 |
| AI2-THOR | 0.57 | 0.59 | 0.69 | 0.56 | 0.56 | 0.62 | 0.79 | 0.63 |
| TDW | 0.73 | 0.69 | 0.86 | 0.7 | 0.67 | 0.89 | 0.81 | 0.76 |

We pre-trained four ResNet-50 models [19] on ImageNet [11], SceneNet [18], AI2-Thor [23] and the TDW-image dataset respectively. We directly downloaded images of ImageNet [11] and SceneNet [18] for model trainings. For a fair comparison, we also created an AI2-THOR dataset with 1.3M images using a controller that captured random images in a scene and classified its segmentation masks from ImageNet synset IDs. We then evaluated the learned representations by fine-tuning on downstream fine-grained image classification tasks using Aircraft [27], Birds [40], CUB [41], Cars [24], Dogs [21], Flowers [31], and Food datasets [8]. We used a ResNet-5- network architecture as a backbone for all the visual perception transfer experiments. For the pre-training, we set the initial learning rate as 0.1 with cosine decay and trained for 100 epochs. We then took the pre-trained weights as initialization and fine-tuned on fine-grained image recognition tasks, using an initial learning rate of 0.01 with cosine decay and training for 10 epochs on the fine-grained image recognition datasets. Table 2 shows that the feature representations learned from TDW-generated images are substantially better than the ones learned from SceneNet [18] or AI2-Thor [23], and have begun to approach the quality of those learned from ImageNet. These experiments suggest that though significant work remains, TDW has taken meaningful steps towards mimicking the use of large-scale real-world datasets in model pre-training. Using a larger transformer architecture [12] with more TDW-generated images might further close the gap with Imagenet pre-trained models on object recognition tasks. We have open-sourced the full image generation codebase to support future research in directions such as this.

**Sound recognition transfer** We also created an audio dataset to test material classification from impact sounds. We recorded 300 sound clips of 5 different materials (cardboard, wood, metal, ceramic, and glass; between 4 and 15 different objects for each material) each struck by a selection of pellets (of wood, plastic, metal; of a range of sizes for each material) dropped from a range of heights between 2 and 75cm. The pellets themselves resonated negligible sound compared to the objects but because each pellet preferentially excited different resonant modes, the impact sounds depend upon the mass and material of the pellets, and the location and force of impact, as well as the material, shape, and size of the resonant objects [39] (more video examples).

Given the variability in other factors, material classification from this dataset is nontrivial. We trained material classification models on simulated audio from both TDW and the sound-20K dataset[47]. We tested their ability to classify object material from the real-world audio. We converted the raw audio waveform to a sound spectrogram representation and fed them to a VGG-16 pre-trained on AudioSet [17]. For the material classification training, we set the initial learning rate as 0.01 with cosine decay and trained for 50 epochs. As shown in Table 3, the model trained on the TDW audio dataset achieves more than 30% better accuracy gains than that trained on the Sound20k dataset. This improvement is plausibly because TDW produces a more diverse range of sounds than Sound20K and prevents the network overfitting to specific features of the synthetic audio set.

Table 3: Sound perception transfer on material recognition.

| Dataset | Accuracy |
|---------|----------|
| Sound-20K | 0.34 |
| TDW | **0.66** |

Table 4: Comparison of the multi-modal physical scene understanding on material and mass classification.

| Method | Material | Mass |
|--------|----------|------|
| Vision only | 0.72 | 0.42 |
| Audio only | 0.92 | 0.78 |
| Vision + Audio | **0.96** | **0.83** |

**Multi-modal physical scene understanding** We used the TDW graphics engine, physics simulation and the sound synthesis technique described in Sec 2.3 to generate videos and impact sounds of objects dropped on flat surfaces (table tops and benches). The surfaces were rendered to have the visual appearance of one of 5 materials. The high degree of variation over object and material appearance, as well as physical properties such as trajectories and elasticity, prevents the network from memorizing features (i.e. that objects bounce more on metal than cardboard). The training and test sets had the same material and mass class categories. However, the test-set videos contained objects, tables, motion patterns, and impact sounds that were different from any video in the training set. Across all videos, the identity, size, initial location, and initial angular momentum of the dropped object were randomized to ensure every video had a unique pattern of motion and bounces. The shape, size, and orientation of the table were randomized, as were the surface texture renderings (e.g., a wooden table could be rendered as "cedar," "pine," "oak," "teak," etc.), to ensure every table appearance was unique. PyImpact uses a random sampling of resonant modes to create an impact sound, such that the impacts in every video had a unique spectro-temporal structure.

For the vision-only baseline, we extracted visual features from each video frame using a ResNet-18 pre-trained on ImageNet, applying an average pooling over 25 video frames to arrive a 2048-d feature vector. For the audio-only baseline, we converted the raw audio waveforms to sound spectrograms and provided them as input for a VGG-16 pre-trained on AudioSet. Each audio-clip was then represented as a 4096-d feature vector. We then took the visual-only features, sound-only features, and the concatenation of visual and sound feature as input to a 2-layer MLP classifier trained for material and mass classification. The results (Table 4) show that audio is more diagnostic than video for both classification tasks, but that the best performance requires audiovisual (i.e. multi-modal) information, underscoring the utility of realistic multi-modal rendering.

## 3.2 Training and Testing Physical Dynamics Understanding

Differentiable forward predictors that mimic human-level intuitive physical understanding have emerged as being of importance for enabling deep-learning based approaches to model-based planning and control applications [25, 4, 29, 15, 5, 9, 2, 36, 13, 14, 32, 46]. While traditional physics engines constructed for computer graphics (such as PhysX and Flex) have made great strides, such routines are often hard-wired, and thus both hard to apply to novel physical situations encountered by real-world robots, and challenging to integrate as components of larger learnable systems. Creating end-to-end differentiable neural networks for intuitive physics prediction is thus an important area of research. However, the quality and scalability of learned physics predictors has been limited, in part by the availability of effective training data. This area has thus afforded a compelling use case for TDW, highlighting its advanced physical simulation capabilities.

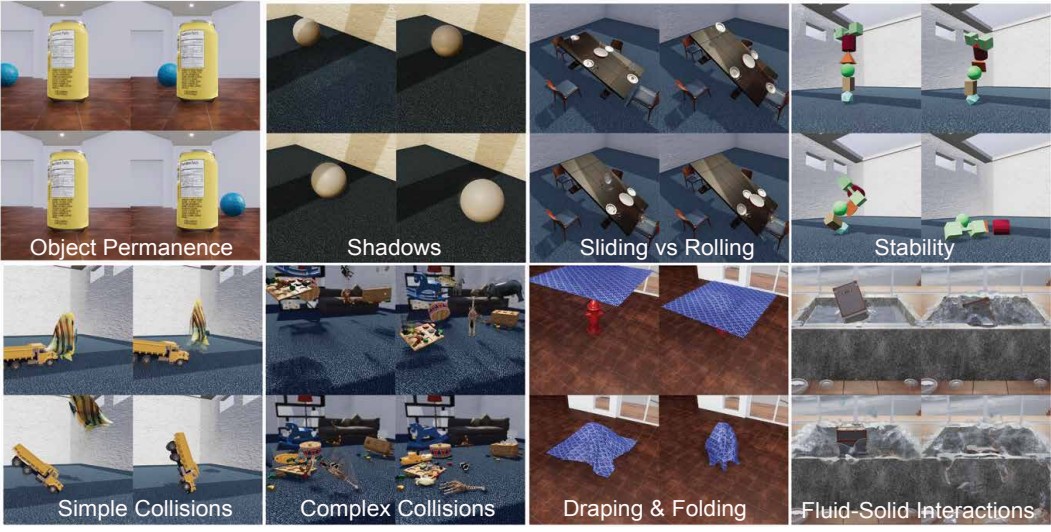

Figure 3: **Advanced Physical Understanding Benchmark.** Scenarios for training and evaluating advanced physical understanding in end-to-end differentiable physics predictors. These are part of a benchmark dataset that will be released along with TDW. Each panel of four images is in order of top-left, top-right, bottom-left, bottom-right ( more video examples).

**Advanced Physical Prediction Benchmark** Using the TDW platform, we have created a comprehensive Pysion benchmark for training and evaluation of physically-realistic forward prediction algorithms [6]. This dataset contains a large and varied collection of physical scene trajectories, including all data from visual, depth, audio, and force sensors, high-level semantic label information for each frame, as well as latent generative parameters and code controllers for all situations. This dataset goes well beyond existing related benchmarks, such as IntPhys [34], providing scenarios with large numbers of complex real-world object geometries, photo-realistic textures, as well as a variety of rigid, soft-body, cloth, and fluid materials. Example scenarios from this dataset are seen in Fig 3 are grouped into subsets highlighting important issues in physical scene understanding, including:
- Object Permanence: Object Permanence is a core feature of human intuitive physics [37], and agents must learn that objects continue to exist when out of sight.

- Shadows: TDW's lighting models allows agents to distinguish both object intrinsic properties (e.g. reflectance, texture) and extrinsic ones (what color it appears), which is key to understanding that appearance can change depending on context, while underlying physical properties do not.
- Sliding vs Rolling: Predicting the difference between an object rolling or sliding – an easy task for adult humans – requires a sophisticated mental model of physics. Agents must understand how object geometry affects motion, plus some rudimentary aspects of friction.
- Stability: Most real-world tasks involve some understanding of object stability and balance. Unlike simulation frameworks where object interactions have predetermined stable outcomes, using TDW agents can learn to understand how geometry and mass distribution are affected by gravity.
- Simple Collisions: Agents must understand how momentum and geometry affects collisions to know that what happens when objects come into contact affects how we interact with them.
- Complex Collisions: Momentum and high resolution object geometry help agents understand that large surfaces, like objects, can take part in collisions but are unlikely to move.
- Draping & Folding: By modeling how cloth and rigid bodies behave differently, TDW allows agents to learn that soft materials are manipulated into different forms depending on what they are in contact with.
- Submerging: Fluid behavior is different than solid object behavior, and interactions where fluid takes on the shape of a container and objects displace fluid are important for many real-world tasks.

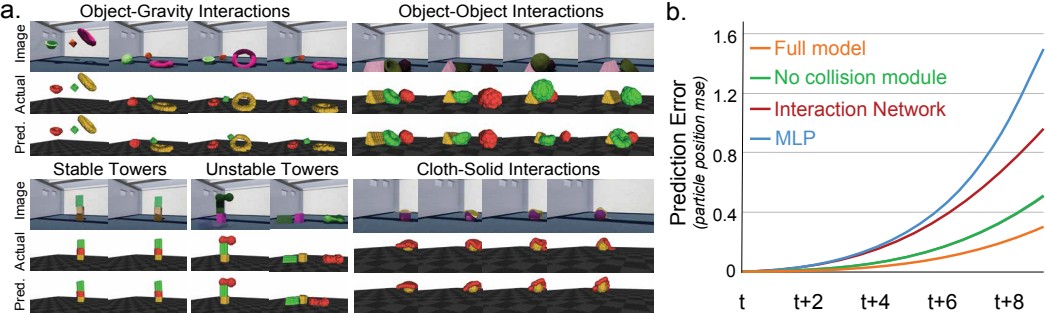

Figure 4: **Training a Learnable Physics Simulator.** (a) Examples of prediction rollouts for a variety of physical scenarios. b) Quantative evaluations of physical predictions over time for HRN compared to no-collision ablation (green), Interaction Network [5] (red), and simple MLP (blue).

**Training a Learnable Intuitive Physics Simulator** The Hierarchical Relation Network (HRN) is a recently-published end-to-end differentiable neural network based on hierarchical graph convolution, that learns to predict physical dynamics in this representation [30]. The HRN relies on a hierarchical part-based object representation that covers a wide variety of types of three-dimensional objects, including both arbitrary rigid geometrical shapes, deformable materials, cloth, and fluids. Here, we train the HRN on large-scale physical data generated by TDW, as a proof of concept for TDW's physical simulation capabilities. Building on the HRN, we also introduce a new Dynamic Recurrent HRN (DRHRN) (**Network Details in Supplement**). that achieves improved physical prediction results that take advantage of the additional power of the TDW dataset generation process.

**Experimental settings** To evaluate HRN and DRHRN accuracy and generalization, we utilize a subset of the scenarios in the advanced physical understanding benchmark. We use objects of different shapes (*bowl*, *cone*, *cube*, *dumbbell*, *octahedron*, *pentagon*, *plane*, *platonic*, *prism*, *ring*, *sphere*) and materials (*cloth*, *rigid*, *soft*) to construct the following scenarios: (1) A **lift subset**, in which objects are lifted and fall back on the ground. (2) A **slide subset**, in which objects are pushed horizontally on a surface under friction. (3) A **collide subset**, in which objects are collided with each other. (4) A **stack subset**, in which objects are (un)stably stacked on top of each other. And (5) a **cloth subset**, in which a cloth is either dropped on one object or placed underneath and lifted up. Three objects are placed in the first four scenarios, as at least three objects are needed to learn indirect object interactions (e.g. stacking). Each subset consists of 256-frame trajectories, 350 for training (~90,000 states) and 40 for testing (~10,000 states).

Given two initial states, each model is trained to predict the next future state(s) at 50 ms intervals. We train models *on all train subsets at once* and evaluate on test subsets separately. We measure the mean-square-error (MSE) between predicted and true particle positions in global and local

Table 5: **Improved Physical Prediction Models.** We measure the global (G) and local (L) position MSE and show qualitative predictions of our **DRHRN** model at *40* time steps in the future on Lift, Slide, Collide, Stack and Cloth data. $|N|$ is the number of objects in the scene.

| [G] ×10⁻¹ [L] ×10⁻² | Lift \|3\| | | Slide \|3\| | | Collide \|3\| | | Stack \|3\| | | Cloth \|2\| | |
|---|---|---|---|---|---|---|---|---|---|---|
| | G | L | G | L | G | L | G | L | G | L |
| **HRN** [30] | 3.27 | 4.18 | 2.04 | 3.89 | 4.08 | 4.34 | 3.50 | 2.94 | 1.33 | 2.22 |
| **DPI** [26] | 3.37 | 4.98 | 3.25 | 3.42 | 4.28 | 4.13 | 3.16 | 2.12 | 0.42 | 0.97 |
| **DRHRN** | **1.86** | **2.45** | **1.29** | **2.36** | **2.45** | **2.98** | **1.90** | **1.83** | **0.24** | **0.64** |

object coordinates. Global MSE quantifies object position correctness. Local MSE assesses how accurately the object shape is predicted. We evaluate predictions 40 frames into the future. For a better visualization of training and test setups, please follow this video link.

**Prediction Results** We first replicate results comparing the HRN against simpler physical prediction baselines. As in the original work, we find that HRN outperforms baseline models without collision-detection or flexible hierarchical scene description (Fig. 4). We then compare DRHRN against strong deterministic physics prediction baselines, including HRN as above, and DPI [26], which uses a different hierarchical message passing order and a hard coded rigid shape preservation constraint. We re-implement both baselines in Tensorflow for direct comparison. Table 5 presents results of the DRHRN comparison. DRHRN clearly outperforms HRN and DPI on all scenarios. It achieves a lower local MSE, indicating better shape preservation which we can indeed observe in the images. All predictions look physically plausible without unnatural deformations (more video results).

## 3.3 Social Agents and Virtual Reality

Social interactions are a critical aspect of human life, but are an area where current approaches in AI and robotics are especially limited. AI agents that model and mimic social behavior, and that learn efficiently from social interactions, are thus an important area for cutting-edge technical development.

**Task Definition** Using the flexibility of TDW's multi-agent API, we have created implementations of a variety of multi-agent interactive settings (Fig. 1f). These include scenarios in which an "observer" agent is placed in a room with multiple inanimate objects, together with several differentially-controlled "actor" agents (Fig. 5a). The actor agents are controlled by either hard-coded or interactive policies implementing behaviors such as object manipulation, chasing and hiding, and motion imitation. Human observers in this setting are simply asked to look at whatever they want, whereas our virtual observer seeks to maximize its ability to predict the behaviors of the actors in this same display, allocating its attention based on a metric of "progress curiosity" [3] that seeks to estimate which observations are most likely to increase the observer's ability to make actor predictions. The main question is whether this form of curiosity-driven learning naturally gives rise to patterns of attention that mirror how humans allocate attention as they explore this same scene for the first time during the experiment.

**Experiments** Intriguingly, in recent work, these socially-curious agents have been shown to outperform a variety of existing alternative curiosity metrics in producing better predictions, both in terms of final performance and substantially reducing the sample complexity required to learn actor behavior patterns [22]. The VR integration in TDW enables humans to directly observe and manipulate objects in responsive virtual environments. Fig. 5 illustrates an experiment investigating the patterns of

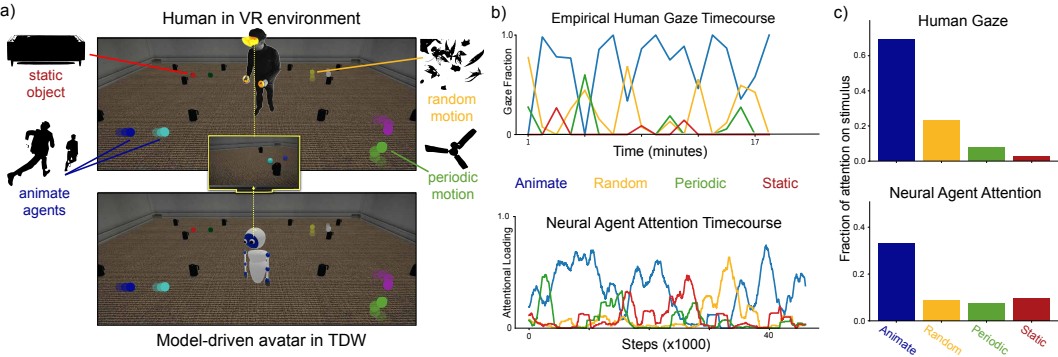

Figure 5: **Multi-Agent and VR Capabilities. a)** Illustration of TDW's VR capabilities in an experiment measuring spontaneous patterns of attention to agents executing spatiotemporal kinematics typical of real-world inanimate and animate agents. By design, the stimuli are devoid of surface features, so that both humans and intrinsically-motivated neural network agents must discover which agents are interesting and thus worth paying attention to, based on the behavior of the actor agents. Example timecourses (panel **b**) and aggregate attention (panel **c**) for different agents, from humans over real time, and from intrinsically-motivated neural network agents over learning time.

attention that human observers exhibit in an environment with multiple animate agents and static objects [20, 16]. Observers wear a GPU-powered Oculus Rift S, while watching a virtual display containing multiple robots. Head movements from the Oculus are mapped to a sensor camera within TDW, and camera images are paired with meta-data about the image-segmented objects, in order to determine which set of robots people are gazing at. Interestingly, the socially-curious neural network agents produce an aggregate attentional gaze pattern that is quite similar to that of human adults measured in the VR environment (Fig. 5b), arising from the agent's discovery of the inherent relative "interestingness" of animacy, without building it in to the network architecture [22]. These results are just one illustration of TDW's extensive VR capabilities in bridging AI and human behaviors.

## 4   Future Directions

We are actively working to develop new capabilities for robotic systems integration and articulatable object interaction for higher-level task planning and execution. **Articulatable Objects.** Currently only a small number of TDW objects are modifiable by user interaction, and we are actively expanding the number of library models that support such behaviors, including containers with lids that open, chests with removable drawers and doors with functional handles. **Humanoid Agents.** Interacting with actionable objects or performing fine-motor control tasks such as solving a jigsaw puzzle requires agents with a fully articulated body and hands. We plan to develop a set of humanoid agent types that fulfill these requirements, with body movement driven by motion capture data and a separate gesture control system for fine motor control of hand and finger articulation. **Robotic Systems Integration.** Building on the modular API layering approach, we envision developing additional "ultra-high-level" API layers to address specific physical interaction scenarios. We are also exploring creating a PyBullet [10] "wrapper" that would allow replicating physics behaviors between systems by converting PyBullet API commands into comparable commands in TDW.

## Acknowledgments and Disclosure of Funding

This work was supported by MIT-IBM Watson AI Lab and its member company Nexplore, ONR MURI, DARPA Machine Common Sense program, ONR (N00014-18-1-2847), Mitsubishi Electric, and NSF Grant BCS-192150.

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
