# OpenReview forum: "ThreeDWorld: A Platform for Interactive Multi-Modal Physical Simulation"
_NeurIPS.cc/2021/Track/Datasets_and_Benchmarks/Round1 — NeurIPS 2021 Datasets and Benchmarks Track (Round 1)_

### Official Review · Reviewer_Dgpm · 2021-07-03
**A one-in-all kind of simulator for embodied learning**

**Rating:** 7
**Confidence:** 3
**Clarity:** The paper is written very clearly.

**Strengths:**

1. Holisticity: AFAIK, this is the only simulation platform that has provisions for solving tasks in 3 main embodied learning paradigms together -- navigation, manipulation and interaction. Although, the extent to which it currently allows manipulation and interaction is kind of limited in comparison to other state-of-the-art simulators (improvements in such directions have been promised as part of future work), such integration could allow for research in building systems through multi-task learning, transfer learning, etc. It also serves as a common platform for comparing different visual or multi-modal tasks and analyze their complexities.
2. Simulation Realism: the level of photo-realism is pretty good and at par with other state-of-the-art simulators that I know of, if not better. The level of realism in physical interactions is also impressive and it tries to cover a wide range of scenarios (hard object vs. hard object, soft material vs. hard object, liquid vs hard object). The realism is also evaluated by visual and audio recognition experiments, where the training is done on simulated data and testing is done on real-world data.
3. User interaction: the simulation also allows for interaction with a human user through VR. This, in my opinion, is an important feature as it allows for the collection of data containing fine-grained human control and manipulation, that's often hard to script. Such data could help in the modeling of human behavior and learning from human demonstrations.

**Weaknesses:**

1. Missing comparison in terms of simulation speed: a very vital feature of a simulation platform is the speed at which the simulator renders input data and allows for feedback control. The comparison of ThreeDWorld with other simulators in terms of rendering speed is missing.

2. Missing comparison with other simulators in terms of realism: although the paper compares different aspects of its simulation with other simulators in a qualitative fashion in table 1, it would be nice to see some numbers. For example, one could perform similar visual and audio recognition experiments in other state-of-the-art simulators (Habitat + SoundSpaces is one example as it provides both audio and visual simulation) and compare the results. It would also be nice to compare the physics simulation with some other state-of-the-art simulator that allows for realistic physical interactions.

P.S.: I am leaning towards acceptance but would like to see the above-listed concerns addressed.

**Additional Feedback:**

None

**Correctness:**

The claims made in the paper and the supporting experiments/evaluation methods look correct to me.

**Documentation:**

The simulation platform looks well-documented.

**Ethics:**

I couldn't spot any ethical issues.

**Relation To Prior Work:**

The work extensively compares with prior work and tries to address their shortcomings. However, some more evaluation would make the work even stronger. I have listed such possible comparisons as part of 'Weakness'.

**Summary And Contributions:**

This work presents a holistic simulation platform to evaluate various embodied tasks. It covers the most important embodied tasks that constitute recent research in embodied learning: navigation, manipulation, and interaction. Besides, it also allows for the rendering of realistic multimodal inputs, i.e. both audio and visual. Additionally, the interactions between the agent and surrounding object and between surrounding objects are made to obey laws of physics, thus adding another dimension of realism to the simulation and allowing for training systems that can reason about and infer such laws of interaction. This is very important for building embodied agents that we can deploy in the real world.


Edit: score updated on July 13th

---

> ### Author Response · Authors · 2021-07-10
> **Response to Reviewer3 (Part 1)**
>
> Thank you very much for your constructive comments and suggestions! Below we address the comments one by one.
>
>
> > Benchmark image rendering speed
>
> - Thanks for your questions. On an Intel i7-7700K@4.2GHz GPU and NVIDIA GeForce GTX 1080, TDW can capture images (render and return image data to the controller) at 380 frames per second (fps). If only object metadata is returned, TDW can achieve 850 fps.  We will include these numbers in the main paper.
>
> - According to its own benchmarks, AI2-THOR for example achieves a maximum of 240 FPS in a similar test; messages in AI2-THOR are sent via HTTP while TDW messages are sent via lower-level TCP sockets, which is much faster.
>
> - More results of image rendering speed of TDW on different hardware could be found here (https://github.com/threedworld-mit/tdw/blob/master/Documentation/benchmark/benchmark.md).
>
> >Quantitative comparisons on visual recognition.
>
> - We have provided one comparison with the closest state-of-the-art alternative platform (AI2THOR) for image recognition accuracy (Table 2). Results with TDW images were significantly better than those for AI2THOR.
>
> - We note that Habitat can generate high-quality images using real-world RGB-D indoor scans but does not provide 3D assets that can be used to generate large-scale image recognition datasets (they are not intended for this purpose since this simulator is mainly designed for indoor embodied navigation).
>
> > Quantitative comparisons on audio recognition.
>
> - We emphasize that there are few other options for audio simulators that we can compare to. We included a comparison with the Sound20k dataset generated by an alternative SOTA audio simulator and the results show that we achieve much better sim2real transfer results.
>
> - Habitat has thus far not released an audio simulator. They instead released a sound-space dataset with pre-calculated room impulse responses.  Moreover, Habitat only supports environmental reverberation, and cannot generate physics-based impact sound, so they cannot create datasets to study inverse physics problems like us. We hope that our platform will stimulate more work in this area.

---

> > ### Comment · Reviewer_Dgpm · 2021-07-13
> > **Reply to changes**
> >
> > All my questions have been adequately addressed. I am willing to increase my rating during the discussion period.

---

> ### Author Response · Authors · 2021-07-10
> **Response to Reviewer3 (Part 2)**
>
> >Quantitative comparisons on physics
>
> There are a number of ways one could compare TDW’s physics simulation capacity to alternative approaches, which we’ll discuss here.
>
> As background, it’s important to note that TDW doesn’t introduce its own underlying physics engine (we realize you probably understand that already). Instead, you can think of our contribution as integrating several hiqh-quality existing real-time physics engines into a framework that can also produce high-quality visual and auditory renderings.  In making this integration, we had three objectives:
> - That the integration is *flexible*.  That is, users could easily set up a wide variety of physical scenarios, placing any type of object in any location in any state, with controllable physical parameters.  This enables us to create physics-related benchmarks with highly variable situations while being able to generate near-photorealistic renderings corresponding to those situations.
> - The actual physics engines had to cover a wide variety of materials, not just (say) rigid-body interactions.  We achieve this aim by seamlessly integrating PhysX (a good rigid-body simulator) and FLeX (a state-of-the-art multi-material simulator for non-rigid and rigid-non-rigid interactions).
> - TDW had to provide a large library of high-quality assets with good physics colliders and realistic rigid and (especially) non-rigid material types, to allow users to take advantage of the power of the physics engines and easily be able to produce interesting and useful scenes.
>
>
>
> Now, returning to R3’s original comparison question.
>
> - One way to compare TDW to alternatives would be to focus on other general-purpose simulation frameworks. However,  we aren’t aware of any other general-purpose simulation framework that meets the three criteria laid out above.  Other competitive simulation frameworks (e.g., Habitat) don’t have full-fledged non-rigid body physics at all, don’t provide the ability to easily initialize novel scenes with arbitrary objects placed within user-designed parameters, or don’t provide a library of non-rigid assets (in particular non-rigid assets). Obviously, Habitat is great for what it was designed for but just isn’t meant for the purposes that TDW is. So it would be hard to even mount a direct comparison, as even setting up the key scenarios for physics comparison (like scenes with cloth-rigid interaction) would just be inaccessible or very hard in (e.g.) Habitat.
>
> - On the other hand, there are other real-time pure-physics engines (e.g. without rendering) that compete with the physics engines TDW integrates, such as Mujoco, PyBullet, and several others. In theory, one could do a comparison of these engines, as pure physics engines, with the ones TDW uses (PhysX and FLeX). But this comparison might not be that informative. On the one hand, on the domains where these engines overlap with competitors like PyBullet -- mostly in simulating rigid-body interactions -- they’d probably all be about equally good (after all, rigid-body simulation has been a solved problem for a while).  On the other hand, on the domains where FLeX excels (e.g. non-rigid interactions), alternatives like PyBullet are much less well-developed, so the comparison would be unfair to the alternatives. This is, of course, the reason we chose to integrate FLeX in the first place!
>
> -Finally, there are frameworks for non-real-time high-fidelity physics simulations, such as (e.g.) differential finite element methods (including DiffTaichi, among others). These are really interesting and have the potential to be better than what we’ve integrated, but they’re not yet general-purpose enough and real-time efficient to quite be candidates for integration into (or comparisons with) what TDW offers.  Hopefully, in the near future, the potential for integrating these super-high-fidelity methods will become realistic options!
>
>
>
> We realize from your comments that some of the above issues could be more clearly articulated in the paper, and we plan to integrate the above points into the final version.

---

### Official Review · Reviewer_ie62 · 2021-07-05
**It’s really an impressive work which could significantly facilitate future research related to a variety of tasks.**

**Rating:** 9
**Confidence:** 3
**Correctness:** The construction process is conducted…
**Clarity:** Yes, it is well written and easy to f…

**Strengths:**

1.	TDW has several unique properties that distinguish it from previous works, including the real-time rendering of both indoor and outdoor scenes, comprehensive physics simulation, diverse user interaction, etc.
2.	This work is in a promising direction, which could largely promote future research towards many tasks, including but not limited to scene understanding, action prediction, multi-agent interactions, robot controlling, etc. All these are significant for building more intelligent AI systems that could better perceive and understand the world.


**Weaknesses:**

The usage seems a little complicated. More demo code and sample cases are encouraged to be included in their repo.

**Additional Feedback:**

Hope this could be an ongoing effort, including more objects, action types, etc, which could better simulate real-world situations.

**Documentation:**

The documentation is clear and sufficient. The codebase is well provided.

**Relation To Prior Work:**

Yes. TDW is not limited to specific domains, it’s a general-purpose platform.


**Summary And Contributions:**

This work proposes ThreeDWorld, a 3D virtual world simulation platform being able to support multi-modal interactions between various agents and diverse objects. TDW could be used to learn visual features, produce synthetic datasets, training intelligent agents, etc.
This general-purpose world simulator has various potential applications and paves the way for diverse tasks.

---

> ### Author Response · Authors · 2021-07-10
> **Response to Reviewer2**
>
> Thanks for your encouraging feedback and valuable suggestions.  We plan to record more hands-on video tutorials about this platform and make it easy for novices. We will also include more objects, action types, and other useful features into this platform as promised in the future work section.

---

### Official Review · Reviewer_xgjC · 2021-07-06
**A virtual simulation framework and a platform for interactive simulation**

**Rating:** 7
**Confidence:** 3
**Clarity:** Yes

**Strengths:**

The paper compiles a great effort in implementing a virtual simulation framework which leverage from existing art. It would likely facilitate progress in the field. The authors collect several datasets and open source them. The work combines several existing simulators as well, so it subsumes those efforts to a certain extent.

**Weaknesses:**

Simulated data are widely used in sciences (e.g. particle physics) and embodied AI. A primary challenge with simulated data is to understand and quantify the potential sources of systematic differences between simulated and real data, and provide ways to correct for these differences to match the state-of-the-art. The authors state the quality of the systems trained on the TWD data have begun to approach the quality of those learned from ImageNet. I think the adoption of this platform depends on whether the agents would be as good as those trained on real data, otherwise it can only be used for experimentation purposes. For instance, would adding more TWD generated data allow to surpass the ImageNet-trained model?

**Additional Feedback:**

Additional quantitative examination of how well feature representations learned using TDW-generated images and audio data transfer to real world scenarios would strengthen the paper.

**Correctness:**

Yes, to the best of my knowledge, but I am not an expert in robotics and embodied AI.

**Documentation:**

Yes

**Ethics:**

No ethics concerns.

**Relation To Prior Work:**

Builds on the prior work.

**Summary And Contributions:**

The paper describes the ThreeDWorld platform for interactive multi-modal physical simulation which allows users to simulate sensory data and mechanical interactions between robot agents and objects in a variety of 3D environments. The paper provides a detailed system overview and comparison of the provided capabilities to the previous physical simulators. It also provides an example downstream application of the ThreeDWorld system, and evaluation of an agent trained on the synthetic image and sound data collected directly using the physical simulation system.

Edit: score updated on July 19th.

---

> ### Author Response · Authors · 2021-07-10
> **Response to Reviewer1**
>
> Thank you very much for your constructive comments and suggestions! Below we address the comments one by one.
>
>
> > Paper Title
>
> Thanks for your suggestions! Our goal is to build a general-purpose simulator that is not tailored for any one application. We indeed find that this simulator has already attracted much attention in the robotics and embodied AI community. On the other hand, researchers in neuroscience and cognitive science also find our simulator exciting and useful, since it allows them to easily create high-fidelity multi-modal datasets as well as to collect human behavioral data to better understand both human and machine perception systems. We also note that the second reviewer noted TDW’s general applicability. Thus we are reluctant to target the paper exclusively at the interactive embodied AI community, even though that is one of the key constituencies for the work.
>
> We can see how “multi-modal physical simulation” could seem a little bit broad, and are open to alternatives. That said, we think the main communities that will care about the paper are likely to be interested in the types of physics that are included in our simulator, so having the word “physics” in the title seems like a useful signal in that respect. We would welcome other suggestions that you might have.
>
>
>
> > Would adding more TDW generated data allow to surpass the ImageNet-trained model?
>
> This is a great question that TDW should enable to be answered. We do agree that using a larger transformer architecture with more TDW generated images might further close the gap with imagenet pre-trained model on image recognition. We have open-sourced the full image generation codebase that can support future research in this direction.
>
> But we respectfully push back on the premise of the question that “the adoption of this platform depends on whether the agents would be as good as those trained on real data”. The most promising applications of TDW are in domains where real-world datasets are nonexistent or very far from being optimized, such that it is difficult to imagine progress without simulators. This seems likely to be true for most applications involving robotics or mobile agents, where the researcher cannot afford to have the robot running wild across a university or company campus during the algorithm development. But we envision that it will also be true for many problems in multi-modal perception, or physical perception, where there are few existing real-world datasets, and where they would be expensive and laborious to obtain. A good simulator should enable progress in these domains even if some fine-tuning is then required for good transfer to real-world conditions. More generally, our whole enterprise represents a bet that simulators will enable progress in AI. It is hard to know for sure how well the resulting systems will eventually transfer to the real-world, but it is a bet that many in the community see as worth making, and TDW will be useful as we all explore what is possible with simulators.
>
> We also emphasize that the purpose of the comparisons with ImageNet-trained models was to provide a very stringent test of the fidelity of TDW’s visual rendering. Object recognition is arguably the most challenging sim2real image recognition task, as there already exists a large high-quality dataset (ImageNet) for visual representation learning. Yet we show that even in this domain we can achieve reasonable performances using TDW-generated images, significantly better than other simulators (e.g. A2THOR).
>
> As for audio recognition, we do not know of any large-scale audio dataset with ground-truth physical parameters. This is partly why TDW seems valuable for this problem. We show in the paper that our simulator can generate large-scale audio datasets for model pre-training on inverse physics problems (e.g. material recognition) and also show that models trained on this dataset can transfer well to the real world. We consider that significant progress in that domain. And we are actively working on improving the range of environmental audio that can be rendered (including sounds of scraping and rolling, and a large range of object materials and shapes).
>
> We emphasize that TDW opens up new opportunities to many under-explored but very fundamental sim2real transfer challenges on physical scene understanding.  In Section 4.2, we show that TDW allows us to generate a first-of-its-kind advanced physical prediction benchmark that enables researchers to train stronger particle-based dynamic models. This is another example where it is extremely hard to collect large-scale real-world data for model training. We believe that using TDW for model pre-training and then fine-tuning on real-world data is a promising direction to make progress in these important areas.
>
>
> Please let us know if you have any other questions!

---

> > ### Comment · Reviewer_xgjC · 2021-07-20
> > **Reply to reviews**
> >
> > My concerns have been addressed by the authors. I have revised my rating accordingly.

---

### Decision · Program_Chairs · 2021-07-26

**Decision:**

Accept

**Comment:**

The proposed platform for Interactive Multi-Modal Physical Simulation shows to be very useful and flexible. It provides a large set of features, and experiments can be performed in several domains of interest for the NeurIPS community. All reviewers have a positive opinion regarding the contribution, and the discussion period between authors and reviewers was shown to be useful.